# Hierarchically-Structured Variational Autoencoders for Long Text Generation

## Abstract

Variational autoencoders (VAEs) have received much attention recently as an end-to-end architecture for text generation. Existing methods primarily focus on synthesizing relatively short sentences (with less than twenty words). In this paper, we propose a novel framework, hierarchically-structured variational autoencoders (*hier*-VAE), for generating long and coherent units of text. To enhance the model's *plan-ahead* ability, intermediate sentence representations are introduced into the generative networks to guide the word-level predictions. To alleviate the typical optimization challenges associated with textual VAEs, we further employ a hierarchy of stochastic layers between the encoder and decoder networks. Extensive experiments are conducted to evaluate the proposed method, where *hier*-VAE is shown to make effective use of the latent codes and achieve lower perplexity relative to language models. Moreover, the generated samples from *hier*-VAE also exhibit superior quality according to both automatic and human evaluations.

## 1 Introduction

Probabilistic *generative* models for text have received considerable attention recently, in part because of their ability to leverage abundant *unlabeled* data and learn text representations with strong generalization abilities. Combined with the flexibility of deep neural networks, generative modeling provides a powerful and effective framework to estimate the underlying probability distribution of *sentences* or *documents*, and to potentially capture the rich sequential information inherent in natural language. In this work, we focus on one specific type of generative model: the variational autoencoder (VAE) (Kingma & Welling, 2013).

The VAE maps a text sequence into a continuous latent variable, or *code*, via an inference (encoder) network; a generative (decoder) network is utilized to reconstruct the input sentence, conditioned on samples from the latent code (via its posterior distribution). The inference and generative networks, parametrized by deep neural networks, can be optimized jointly by maximizing a

| |
|---|
| i went there a few days ago and was told it was horrible , i was told there was a horrible customer service , and they were rude, i would not recommend this location to anyone , i would not recommend this location to anyone . |

Table 1: Sentences generated from a baseline text-VAE model, as described in (Bowman et al., 2016).

variational lower bound of the marginal distribution of the training corpus. The advantages of leveraging the neural variational inference (NVI) framework have been demonstrated on a wide range of natural language processing (NLP) tasks (Bowman et al., 2016; Miao et al., 2016; Yang et al., 2017; Serban et al., 2017; Shah & Barber, 2018; Kim et al., 2018; Kaiser et al., 2018; Bahuleyan et al., 2018; Deng et al., 2018). Although NVI-based methods have demonstrated success in modeling (generating) relatively short sentences (typically less than twenty words), there are several challenges that need to be addressed to facilitate *long* text generation with deep latent-variable models.

The first challenge originates from the nature of recurrent neural networks (RNNs), which have become the cornerstone for text generation. RNNs are typically trained with gradient descent, using backpropagation through time (BPTT) (Rumelhart et al., 1986). However, it is very difficult to scale such a training paradigm to long sequences, due to vanishing or exploding gradients (Hochreiter & Schmidhuber, 1997). Moreover, the information of the entire generated sequence needs to be stored in the RNN intermediate hidden states, which could be very demanding for RNNs with the increasing length of generated texts. As shown in Table 1, the sentence sampled from a vanilla text-VAE model (Bowman et al., 2016) tends to repeat itself as the sequence proceeds, leading to inferior

generation results. In addition to the issue of repetitiveness, to generate globally-coherent long text sequences, it is desirable that both the *higher-level* abstract features and *lower-level* concrete details (*e.g.*, specific word choices) of longer texts can be leveraged by the generative network. This intuition is also hard to capture with a single-layer (flat) RNN-based generative network.

Another underlying challenge of generating *long* text relates to the "*posterior collapse*" issue inherent in training variational autoencoders with a strong *autoregressive* decoder (Bowman et al., 2016; Yang et al., 2017; Semeniuta et al., 2017; Shen et al., 2018). Bowman et al. (2016) found that while employing an LSTM-based VAE for text, the posterior distribution of latent codes tends to match the prior distribution regardless of the input sequence (the KL divergence between the two distributions is very close to zero). Consequently, the information from the latent variable is not leveraged by the generative network (Bowman et al., 2016). To mitigate this issue, several strategies have been proposed (see optimization challenges in Section 2) to make the decoder less autoregressive; thus, less *contextual information* is utilized by the decoder network (Yang et al., 2017; Shen et al., 2018). However, reducing the autoregressive power of the decoder to make the latent variable more informative can be suboptimal from a text-generation perspective, since the coherence of the generated paragraphs may be sacrificed.

Motivated by these observations, we propose a hierarchically-structured variational autoencoder (*hier*-VAE), a novel variational approach for generating long sequences of text. To improve the model's plan-ahead capability for capturing long-term dependency, our approach exploits a hierarchical LSTM decoder as the generative network. Specifically, a set of intermediate sentence-level representations are introduced during the decoding process (via a sentence-level LSTM network), providing additional semantic information while making *word-level* predictions.

To ameliorate the *posterior collapse* issue associated with training a VAE for text, we further propose leveraging a hierarchy of latent variables between the convolutional inference networks and recurrent generative networks. With multiple stochastic layers, where the prior of each latent variable is inferred from data, rather than fixed (as in a standard VAE setting (Kingma & Welling, 2013)), the sampled *latent codes* at the bottom level are endowed with more flexibility to abstract meaningful features from the input sequences. We evaluate the proposed *hier*-VAE comprehensively on language modeling, generic (unconditional) text generation, and conditional generation tasks. The proposed model demonstrates substantial improvement relative to several baseline methods, in terms of *perplexity* on language modeling and quality of generated samples (measured by both the BLEU statistics and human evaluation). We further generalize our method to the conditional generation scenario to demonstrate the versatility of the proposed method.

## 2 RELATED WORK

**VAE for text generation**   The variational autoencoder, trained under the neural variational inference (NVI) framework, has been widely used for generating text sequences (Bowman et al., 2016; Yang et al., 2017; Semeniuta et al., 2017; Zhao et al., 2017). By encouraging the latent feature space to match a prior distribution within an encoder-decoder architecture, the learned latent variable could potentially encode high-level semantic features and serve as a global representation during the decoding process (Bowman et al., 2016). The generated results are also endowed with better diversity due to the sampling procedure of the latent codes (Zhao et al., 2017). However, existing works have mostly focused on generating one sentence (or multiple sentences with at most twenty words in total). The task of generating relatively longer units of text has been less explored. Our proposed architecture seeks to extend the NVI framework for long-form text generation by exploiting the hierarchical nature of a paragraph (multi-sentence text sequences), where the outputs from a sentence-level LSTM are utilized to guide word-level predictions.

**Optimization challenges with text-VAEs**   The "posterior collapsing" issue associated with training text-VAEs was first outlined by (Bowman et al., 2016), where they have utilized *KL divergence annealing* and *word dropout* strategies to mitigate the optimization challenge. However, their resulting method still does not outperform a pure language model in terms of the final perplexity. Yang et al. (2017) argue that the small KL term relates to the strong autoregressive nature of an LSTM generative network, and they proposed to utilize a dilated CNN as a decoder to improve the informativeness of the latent variable. (Zhao et al., 2018b) proposed to augment the VAE training objective with an additional mutual information term. Unfortunately, such a strategy results in an intractable integral in the case where the latent variables are continuous. Our work tries to tackle the "posterior

collapse" issue from two perspectives: 1) more flexible priors are assumed over the latent variables (learned from the data); 2) the hierarchical structure within a paragraph is taken into account, so that the latent variables can focus less on the local information (*e.g.*, word-level coherence) but more on the global features.

**Hierarchical structures in NLP** Natural language is inherently organized in a hierarchical manner (characters form a word, words form a sentence, sentences form a paragraph, paragraphs from a document, *etc.*). Yang et al. (2016) employed hierarchical LSTM encoders at the word- and sentence-level along with an attention mechanism to learn document representations. Li et al. (2015) proposed a hierarchical autoencoder to reconstruct long paragraph of texts. One method conceptually similar to our work is that of (Serban et al., 2017), which produces a stochastic latent variable for each sentence during decoding. In contrast, our model encodes the entire paragraph into one *single* latent variable. As a result, the latent variable learned in our model relates more to the global semantic information of a paragraph, whereas those in (Serban et al., 2017) mainly contain the local information of a specific sentence. Therefore, their model is not suitable for tasks such as latent space interpolation. Our approach bears close resemblance to the VHCR model (Park et al., 2018). However, there exist several key differences that make our work unique: *i*) both latent variables in our *hier*-VAE are designed to contain global information. More importantly, although the local/utterance variables are generated from the global latent variable in VHCR, the priors for the two sets of latent variables are both fixed as standard diagonal-covariance Gaussian. In contrast, the prior of the bottom-level latent variable in our model is learned from the data (thus more flexible relative to a fixed prior), which exhibits promising results in terms of mitigating the "posterior collapse" issue (see Table 2); *ii*) the underlying data distribution of the entire paragraph is captured in the bottom-level latent variable in *hier*-VAE. While in the setup of VHCR, the responses are modeled/generated conditioned on both the latent variables and the contexts. Therefore, the (global) latent variable learned by our model should contain more information. Modeling hierarchical structures has been shown to be beneficial for generating others types of data as well, such as images (Sønderby et al., 2016; Gulrajani et al., 2016), music (Roberts et al., 2018), or speech (Hsu et al., 2017).

## 3 HIERARCHICALLY-STRUCTURED VAE MODEL

Our goal is to build a model that can generate human-generated-like sentences conditioned on random samples from the (prior) latent space. The proposed method, as illustrated in Figure 1, is built upon the VAE framework introduced in (Bowman et al., 2016), and consists of a hierarchical CNN *inference* network (encoder) and a hierarchical LSTM *generative* network (decoder). The encoder attempts to extract features at different semantic levels, which are then leveraged by the decoder to allow long-term planning while synthesizing long-form text. We define long-form text as sequences of sentences (such as a Yelp review on a business, or an abstract of an article) that are arranged in a logical manner and usually follow a definite plan for development.

### 3.1 VARIATIONAL AUTOENCODER

Recurrent neural network (RNN) language models (Mikolov et al., 2011), which predict each token conditioned on the entire history of previously generated tokens, have been widely employed for unsupervised generative modeling of natural language. Although effective, they typically do not explicitly capture the *global* (*high-level*) features (*e.g.*, topic or style properties) of a paragraph or sentence. Motivated by this, the variational autoencoder (VAE) was introduced by (Bowman et al., 2016) as an alternative for generative text modeling, where a stochastic latent variable is augmented as the additional information to modulate the sequential generation process. Let $x$ denote a text sequence, which consists of $L$ tokens, *i.e.*, $x_1, x_2, ..., x_L$. A continuous latent code $z$ is first sampled from the prior distribution $p(z)$ (a multivariate Gaussian prior is typically chosen), and then the text sequence $x$ is generated, conditioned on $z$, via a generative (decoder) network. Therefore, the generating distribution can be written as:

$$p_{\boldsymbol{\theta}}(\boldsymbol{x}|\boldsymbol{z}) = p_{\boldsymbol{\theta}}(x_1|\boldsymbol{z}) \prod_{t=2}^{L} p_{\boldsymbol{\theta}}(x_t|x_1, x_2, ..., x_{t-1}, \boldsymbol{z}) \tag{1}$$

where $\boldsymbol{\theta}$ are the parameters of the generative network (to be learned). Since the information from latent variable $z$ is incorporated during the entire generation process, the high-level properties of the corresponding paragraph or sentence could be better leveraged (Bowman et al., 2016; Yang et al., 2017). To estimate the parameters $\boldsymbol{\theta}$, one would ideally maximize the marginal distribution

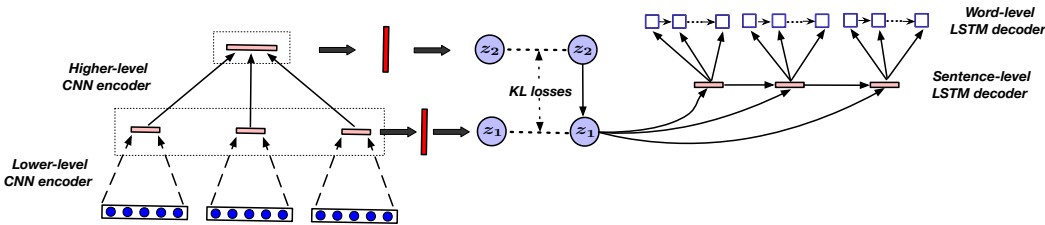

Figure 1: Schematic diagram of the proposed *hierarchically-structured* VAE.

$p(\boldsymbol{x}) = \int p(\boldsymbol{z})p(\boldsymbol{x}|\boldsymbol{z})d\boldsymbol{z}$. However, computing this marginal is intractable in most cases of practical interest. Instead, we maximize a variational lower bound, as typically employed in the VAE framework (Kingma & Welling, 2013): $\mathcal{L}_{\text{vae}} = \mathbb{E}_{q_\phi(\boldsymbol{z}|\boldsymbol{x})}[\log p_{\boldsymbol{\theta}}(\boldsymbol{x}|\boldsymbol{z})] - D_{KL}(q_\phi(\boldsymbol{z}|\boldsymbol{x})||p(\boldsymbol{z}))$. Note that $q_\phi(\boldsymbol{z}|\boldsymbol{x})$ is employed to approximate the true posterior, which is usually called the inference or encoder network (parameterized by $\phi$). The Kullback-Leibler (KL) divergence term $D_{KL}(q_\phi(\boldsymbol{z}|\boldsymbol{x})||p(\boldsymbol{z}))$, which can be written in closed-form (Kingma & Welling, 2013), encourages the approximate posterior distribution $q_\phi(\boldsymbol{z}|\boldsymbol{x})$ to be close to the prior $p(\boldsymbol{z})$. Note that the VAE prior is often taken to be a diagonal-covariance Gaussian for convenience. Although the NVI framework has been shown effective for various NLP tasks, directly applying it to *long text generation* is still challenging. The reasons are two-fold: (1) the lack of a long-term planning mechanism, which is critical for generating *semantically-coherent* long texts (Serdyuk et al., 2017); (2) the *posterior collapse* issue. Concerning (2), it was demonstrated in (Bowman et al., 2016) that due to the autoregressive nature of the RNN, the decoder tends to ignore the information from $\boldsymbol{z}$ entirely, resulting in an extremely small KL term.

## 3.2 HIERARCHICAL LSTM-BASED GENERATIVE NETWORK

To improve the *plan-ahead* ability of the generative network, we introduce intermediate sentence representations to facilitate the sequential generation process. Instead of directly making word-level predictions merely conditioned on the semantic information from $\boldsymbol{z}$, a series of *plan vectors* are first generated based upon $\boldsymbol{z}$ with a *sentence-level* LSTM decoder. The hypothesis underlying this design is that explicitly incorporating the knowledge of (inherently hierarchical) paragraph structure is beneficial for the model to capture sentence-level coherence and potentially mitigate repetitiveness. Intuitively, while predicting each token, the information from both previously generated words and sentence-level representations can be utilized (see Figure 1).

Assume that the input paragraph consists of $M$ sentences, and the $t$-th sentence contains $N_t$ words (for $t = 1, 2, ..., M$). To generate the plan vectors, the sampled latent code $\boldsymbol{z}$ is first sent through a one-layer multi-layered perceptron (MLP), with ReLU activation functions, to obtain the starting state of the sentence-level LSTM decoder. Subsequently, the representations for each sentence, the *plan vectors* are inferred in a sequential manner: $\boldsymbol{h}_t^s = \text{LSTM}^{\text{sent}}(\boldsymbol{h}_{t-1}^s, \boldsymbol{z})$, for $t = 1, 2, 3, ..., M$. Note that $\boldsymbol{z}$ is fed to the higher-level LSTM at every time step to predict the sentence representations. This is because the latent code can be regarded as a paragraph-level abstraction, and thus is informative for determining the semantics of each generated subsequence.

The generated sentence-level plan vectors are then passed through a word-level LSTM decoder to generate the words for each sentence. Similar to the sentence-level decoder, the corresponding *plan vector*, $\boldsymbol{h}_t^s$, is concatenated with the word embedding of the previous word and fed to the LSTM at every time step (see Figure 1). Let $w_{t,i}$ denote the $i$-th token in the $t$-th sentence (teacher forcing is employed during training, and we use greedy decoding at test time). This process can be expressed as (for $t = 1, 2, ..., M$ and $i = 1, 2, 3, ..., N_t$):

$$\boldsymbol{h}_{t,i}^w = \text{LSTM}^{\text{word}}(\boldsymbol{h}_{t,i-1}^w, \boldsymbol{h}_t^s, \boldsymbol{W_e}[w_{t,i-1}]), \tag{2}$$

$$p(w_{t,i}|w_{t,<i}, \boldsymbol{h}_t^s) = \text{softmax}(\boldsymbol{V}\boldsymbol{h}_{t,i}^w), \tag{3}$$

$\boldsymbol{h}_{t,0}^w$ is inferred from the corresponding plan vector via an MLP layer. Weight matrix $\boldsymbol{V}$ is used for computing a distribution over words, and $\boldsymbol{W_e}$ are word embeddings to be learned. For each sentence, once the special ⎽END token is generated, the word-level LSTM stops decoding (note that each sentence is padded with an ⎽END token at the preprocessing step). The parameters of the word-level decoder for each sentence are shared, and are trained jointly with the parameters of sentence-level LSTM.

Inspired by Chen et al. (2016), we hypothesize that with the hierarchical decoder networks described above, the latent codes are encouraged to abstract more global and high-level semantic information from the paragraph. While relatively more detailed information (*e.g.*, word-level (local) coherence) can be captured via the word- and sentence-level LSTM networks, the latent variables can focus more on the global information (*e.g.*, paragraph-level semantic features) (Chen et al., 2016). Empirically, we also found that this strategy helps the generative network to make more effective use of the latent variable $z$ (indicated by a larger KL loss term).

### 3.3 HIERARCHICALLY-STRUCTURED LATENT VARIABLES

Due to the use of an autoregressive LSTM decoder for word-level predictions, the problem of "posterior collapse", where the model tends to ignore the latent codes while decoding, may still exist. In this work, we consider one effective way to mitigate this problem by making less restrictive assumptions regarding the prior distribution of the latent variable. Specifically, we propose to leverage a hierarchy of latent variables for text modeling. As shown in Figure 1, the inference network goes upward through each latent variable to infer their posterior distributions, while the generative network samples downward to obtain the prior distributions over the latent variables. In this manner, the prior distribution of latent variable at the bottom is inferred from the top-layer latent codes, rather than fixed (as in a standard VAE model). Consequently, the model is endowed with more flexibility to abstract useful high-level features (Gulrajani et al., 2016), which can then be leveraged by the hierarchical LSTM network. Without loss of generality, here we choose to employ a two-layer hierarchy of latent variables, where the bottom and top ones are denoted as $z_1$ and $z_2$, respectively. Note that this framework can be extended easily to more latent-variable layers. Empirically, we found that the two-layer setup already gives rise to much more informative latent codes, relative to a standard VAE with one stochastic layer.

The posterior distributions over the latent variables are assumed to be conditionally independent given the input $x$. Thus, the joint posterior distribution of the two latent variables can be written as:

$$q_\phi(z_1, z_2|x) = q_\phi(z_2|x)q_\phi(z_1|x) \tag{4}$$

Concerning the generative network, the latent variable at the bottom is sampled conditioning on the one at the top. Thus, we have:

$$p_\theta(z_1, z_2) = p_\theta(z_2)p_\theta(z_1|z_2) \tag{5}$$

To optimize the parameters of the inference and generative networks, the second term in the VAE objective, $D_{KL}(q_\phi(z|x)||p(z))$, can be regarded as the KL divergence between the joint posterior and prior distributions of the two latent variables. Thus, the variational lower bound can be written as (given the assumptions in (4) and (5)):

$$\mathcal{L}_{\text{vae}} = \mathbb{E}_{q_\phi(z_1|x)}[\log p_\theta(x|z_1)] - D_{KL}(q_\phi(z_1, z_2|x)||p_\theta(z_1, z_2)) \tag{6}$$

$$D_{KL} = \int_{z_1, z_2} q_\phi(z_2|x)q_\phi(z_1|x) \log \frac{q_\phi(z_2|x)q_\phi(z_1|x)}{p_\theta(z_2)p_\theta(z_1|z_2)} dz_1 dz_2$$

$$= \int_{z_1, z_2} [q_\phi(z_2|x)q_\phi(z_1|x) \log \frac{q_\phi(z_1|x)}{p_\theta(z_1|z_2)} + q_\phi(z_2|x)q_\phi(z_1|x) \log \frac{q_\phi(z_2|x)}{p_\theta(z_2)}] dz_1 dz_2$$

$$= \mathbb{E}_{q_\phi(z_2|x)}[D_{KL}(q_\phi(z_1|x)||p_\theta(z_1|z_2))] + D_{KL}(q_\phi(z_2|x)||p_\theta(z_2)) \tag{7}$$

Note that the left-hand side of (7) is the abbreviation of $D_{KL}(q_\phi(z_1, z_2|x)||p(z_1, z_2))$. Given the Gaussian assumption for both the prior and posterior distributions, both KL divergence terms can be written in closed-form.

## 4 EXPERIMENTS

### 4.1 EXPERIMENTAL SETUPS

**Datasets** We evaluate the proposed hierarchically-structured VAE on both generic (unconditional) long text generation and conditional paragraph generation (with additional text input as auxiliary information). For the former, we employ two datasets: Yelp Reviews (Zhang et al., 2015) and ArXiv Abstracts (Celikyilmaz et al., 2018). For the conditional-generation case, we consider the task of synthesizing a paper abstract (which typically includes several sentences) conditioned on

the paper title (with the ArXiv Abstract dataset). We employ a hierarchical CNN architecture as the inference/encoder network. Specifically, a sentence-level CNN encoder is first applied to each sentence to obtain a fixed-length vector. Afterwards, a paragraph-level CNN encoder is utilized to aggregate the vectors with respect to all sentences. Details of the dataset, the experimental setup and model architectures are provided in the Supplementary Materials (SM).

**Model specification** To abstract meaningful representations from the input sentences, we employ a hierarchical CNN architecture as the inference/encoder network. Specifically, a sentence-level CNN encoder is first applied to each sentence to obtain a fixed-length vector. Afterwards, a paragraph-level CNN encoder is utilized to aggregate the vectors with respect to all sentences. In the single-variable *hier*-VAE, the paragraph feature vector is fed into linear layers to infer the mean and variance of the latent variable $z$. For the double-variable case, the feature vector is further transformed with two MLP layers, and then is used to compute the mean and variance of the top-level latent variable. The dimension of latent variable $z$ is set to $300$. The dimensions of both the sentence-level and word-level LSTM decoders are set to $512$. For the generative networks, to infer the bottom-level latent variable (*i.e.*, modeling $p(z_1|z_2)$), we first feed the sampled latent codes from $z_2$ to two MLP layers, which is followed by two linear transformation to infer the mean and variance of $z_1$, respectively.

The model is trained using Adam (Kingma & Ba, 2014) with a learning rate of $3 \times 10^{-4}$ for all parameters, with a decay rate of 0.99 for every 3000 iterations. Dropout (Srivastava et al., 2014) is employed on both word embedding and latent variable layers, with rates selected from {0.3, 0.5, 0.8} on the validation set. We set the mini-batch size to 128. Following (Bowman et al., 2016) we adopt the KL cost annealing strategy to stabilize training: the KL cost term is increased linearly to 1 until 10,000 iterations. All experiments are implemented in Tensorflow (Abadi et al., 2016), using one NVIDIA GeForce GTX TITAN X GPU with 12GB memory.

**Baselines** For the language modeling experiments, we implement several baselines: language model with a flat LSTM decoder (*flat*-LM), VAE with a flat LSTM decoder (*flat*-VAE), language model with a hierarchical LSTM decoder (*hier*-LM). For generic text generation, we further consider two recently proposed generative models as the baselines: Adversarial Autoencoders (AAE) (Makhzani et al., 2015) and Adversarially-Regularized Autoencoders (ARAE) (Zhao et al., 2018a). Instead of penalizing the KL divergence term, AAE introduces a discriminator network to match the prior and posterior distributions of the latent variable. Further, AARE proposes to learn a more flexible prior by assuming no specific forms of the latent space. For the proposed hierarchically-structured VAE, we implement two variants, one with a single latent variable and another with double latent variables, denoted as *hier*-VAE-S and *hier*-VAE-D, respectively. Our code will be released to encourage future research.

## 4.2 LANGUAGE MODELING RESULTS

We first evaluate our method on language modeling task using Yelp and ArXiv datasets, where we report the negative log likelihood (NLL) and perplexity (PPL). Following (Bowman et al., 2016; Yang et al., 2017; Kim et al., 2018), we utilize the KL loss term to measure the extent of "posterior collapse". For this experiment we use the following baselines: language model with a flat LSTM decoder, VAE with a flat LSTM decoder, language model with a hierarchical LSTM decoder, VAE with a hierarchical LSTM decoder. As shown in Figure 2, on the Yelp dataset, the standard textVAE with a flat LSTM decoder has a KL divergence term very close to zero, indicating that the generative model makes neligible use of the information from latent variable $\boldsymbol{z}$. Consequently, the *flat*-VAE model obtains slightly worse NNL and PPL relative to a flat LSTM-based language model. In contrast, with a hierarchical LSTM decoder, the KL divergence cost term becomes much larger, demonstrating that the VAE model tends to leverage more information from the latent variable in the decoding stage. Further, the PPL of *hier*-VAE is also decreased from 47.9 to 46.6 (compared with a LM with hierarchical decoder), further showing that the sampled latent codes has indeed helped make word-level predictions.

Moreover, with a hierarchy of latent variables, the *hier*-VAE-D model exhibits an even larger KL divergence cost term (increased from 3.6 to 6.8) than that with a single latent variable, indicating that more information from the latent variable has been utilized by the generative network. This may be attributed to the fact that the latent variable priors of the *hier*-VAE-D model are inferred from the data, rather than a fixed standard Gaussian distribution. As a result, the model is endowed with more flexibility to encode informative semantic features in the latent variables, yet matching their

| *hier*-VAE | *flat*-VAE |
|---|---|
| i have been going to this nail salon for over a year now , the last time i went there , the stylist was nice , but the lady who did my nails , she was very rude and did not have the best nail color i once had . | the staff is very friendly and helpful , **the only reason i** can t give them 5 stars , **the only reason i** am giving the ticket is because of the ticket , **can t help but** the staff is so friendly and helpful , **can t help but** the parking lot is just the same . |
| i am a huge fan of this place , my husband and i were looking for a place to get some good music , this place was a little bit pricey , but i was very happy with the service , the staff was friendly . | i went here for a grooming and a dog , it was very good , **the owner is very nice and friendly , the owner is really nice and friendly** , i don t know what they are doing . |

Table 3: Samples randomly generated from *hier*-VAE-D and *flat*-VAE, which are both trained on the Yelp review dataset. The repetitive patterns within the generated reviews are highlighted. More examples can be found in the supplementary material (Table 14).

posterior distributions to the corresponding priors. More importantly, by effectively exploiting the sampled latent codes, *hier*-VAE-D achieves the best PPL results on both datasets (on ArXiv dataset our hierarchical decoder outperforms the LM by reducing the PPL from $58.1$ down to $54.3$).

## 4.3 GENERIC TEXT GENERATION

We further evaluate our method by examining the quality of generated paragraphs. In this regard, we randomly sample 1000 latent codes and obtain the generated text sequences by feeding them to the trained generative networks. We employ corpus-level BLEU score (Papineni et al., 2002)

| Model | Yelp | | | ArXiv | | |
|---|---|---|---|---|---|---|
| | NLL | KL | PPL | NLL | KL | PPL |
| *flat*-LM | 162.6 | - | 48.0 | 218.7 | - | 57.6 |
| *flat*-VAE | ≤ 163.1 | 0.01 | ≤ 49.2 | ≤ 219.5 | 0.01 | ≤ 58.4 |
| *hier*-LM | 162.4 | - | 47.9 | 219.3 | - | 58.1 |
| *hier*-VAE-S | ≤ 160.8 | 3.6 | ≤ 46.6 | ≤ 216.8 | 5.3 | ≤ 55.6 |
| *hier*-VAE-D | ≤ **160.2** | 6.8 | ≤ **45.8** | ≤ **215.6** | 12.7 | ≤ **54.3** |

Table 2: Results on text modeling for both the Yelp and ArXiv datasets.

to quantitatively evaluate the generated paragraphs. Specifically, we follow the strategy in (Yu et al., 2017; Zhang et al., 2017) and use the entire test set as the reference. The BLEU scores for the 1000 generated sentences are averaged to obtain the final score for each model. As shown in Table 5, VAE tends to be a stronger baseline for paragraph generation, exhibiting higher corpus-level BLEU scores than both AAE and ARAE. This observation is consistent with the results shown in (Cífka et al., 2018). The VAE with hierarchical decoder again demonstrates better BLEU scores than the one with a flat decoder, indicating that the plan-ahead mechanism associated with the hierarchical decoding process indeed benefits the sampling quality. Moreover, *hier*-VAE-D exhibits slightly better results than *hier*-VAE-S. We attribute this to the more flexible prior distribution of *hier*-VAE-D, which improves the ability of the inference networks to extract semantic features from a paragraph and thus yields more informative latent codes.

Table 3 shows an example generated text from *hier*-VAE-H and *flat*-VAE-H. Compared to the our hierarchical model, using flat decoders with flat VAE architecture exibits n-gram repetitions as well as less variations in the descriptions. The hieararhical model, on the other hand, contains more information with less repetitions (word or semantic semantic repetitions) indicating a more coherent generated text.

**The Continuity of Latent Space**  Following (Bowman et al., 2016), we further measure the continuity of the learned latent space. Specifically, two points are randomly sampled from the prior latent space (denoted as $A$ and $B$, respectively). Sentences are generated based on the equidistant intermediate points along the linear trajectory between $A$ and $B$. As shown in Table 4, these intermediate samples are all realistic-looking reviews that are syntactically and semantically reasonable, demonstrating the smoothness of the learned VAE latent space. Interestingly, we even observe that the generated sentences gradually transit from positive to negative sentiment along the linear trajectory. To validate that the sentences are not generated by simply retrieving the training data, we further find the closest instance, among the entire training set, for each generated review. We demonstrate the details of the results in the supplementary material (Table 15).

**Diversity of Generated Paragraphs**  We also evaluate the diversity of random samples from a trained model, since one model might generate realistic-looking sentences while suffering from severe mode collapse (*i.e.*, low diversity). Three metrics are employed to measure the diversity of generated paragraphs: Self-BLEU scores (Zhu et al., 2018), unique $n$-grams (Fedus et al., 2018) and entropy score (Zhang et al., 2018). Specifically, for a set of sampled sentences, the Self-BLEU

| Model | Yelp | | | | ArXiv | | | |
|---|---|---|---|---|---|---|---|---|
| | **BLEU-2** | **BLEU-3** | **BLEU-4** | **BLEU-5** | **BLEU-2** | **BLEU-3** | **BLEU-4** | **BLEU-5** |
| *ARAE* | 0.684 | 0.524 | 0.350 | 0.104 | 0.624 | 0.475 | 0.305 | 0.124 |
| *AAE* | 0.735 | 0.623 | 0.383 | 0.167 | 0.729 | 0.564 | 0.342 | 0.153 |
| *flat*-VAE | 0.855 | 0.705 | 0.515 | 0.330 | 0.784 | 0.625 | 0.421 | 0.247 |
| *hier*-VAE-S | 0.901 | 0.744 | 0.531 | 0.336 | 0.821 | **0.663** | 0.447 | 0.273 |
| *hier*-VAE-D | **0.912** | **0.755** | **0.549** | **0.347** | **0.825** | 0.657 | **0.460** | **0.282** |

Table 5: Evaluation results for generated sequences by our models and baselines on corpus-level BLEU scores.

| Model | BLEU-2 | BLEU-3 | BLEU-4 | Bigrams% | Trigrams% | Quadgrams% | Etp-2 |
|---|---|---|---|---|---|---|---|
| ARAE | 0.725 | 0.544 | 0.402 | 36.2 | 59.7 | 75.8 | 7.551 |
| AAE | 0.831 | 0.672 | 0.483 | 33.2 | 57.5 | 71.4 | 6.767 |
| *flat*-VAE | 0.872 | 0.755 | 0.617 | 23.7 | 48.2 | 69.0 | 6.793 |
| *hier*-VAE-S | 0.865 | 0.734 | 0.591 | 28.7 | 50.4 | 70.7 | 6.843 |
| *hier*-VAE-D | 0.851 | 0.723 | 0.579 | 30.5 | 53.2 | 72.6 | 6.926 |

Table 6: The *self-BLEU scores*, *unique n-gram percentages* and 2-*gram entropy score* of 1000 generated sentences. Models are trained on the Yelp Reviews dataset to evaluate the diversity of generated samples.

metric calculates the BLEU score of each sample with respect to all other samples as the reference (the numbers over all samples are then averaged); the unique score computes the percentage of *unique* $n$-grams within all the generated reviews; and the entropy score measures how evenly the empirical $n$-gram distribution is for a given sentence, which does not depend on the size of testing data, as opposed to unique scores. Note that all three metrics are the lower, the better.

Here, we randomly sample 1000 reviews from each model, and the corresponding results are shown in Table 6. Note that a small self-BLEU score must be accompanied with a large BLEU score to justify the effectiveness of a model, *i.e.*, being able to generate realistic-looking as well as diverse samples. Among all the VAE variants, *hier*-VAE-D shows the smallest BLEU score and largest unique $n$-grams percentage, further demonstrating the advantages of making both the generative networks and latent variables hierarchical. As for AAE and ARAE, although they exhibit better diversity according to both metrics, their corpus-level BLEU scores are much worse relative to *hier*-VAE-D. Thus, we leverage human evaluation for further comparison.

| | |
|---|---|
| A | the service was great, the receptionist was very friendly and the place was clean, we waited for a while, and then our room was ready . |
| • | same with all the other reviews, this place is a good place to eat, i came here with a group of friends for a birthday dinner, we were hungry and decided to try it, we were seated promptly. |
| • | this place is a little bit of a drive from the strip, my husband and i were looking for a place to eat, all the food was good, the only thing i didn t like was the sweet potato fries. |
| • | this is not a good place to go, the guy at the front desk was rude and unprofessional, it s a very small room, and the place was not clean. |
| • | service was poor, the food is terrible, when i asked for a refill on my drink, no one even acknowledged me, they are so rude and unprofessional. |
| B | how is this place still in business, the staff is rude, no one knows what they are doing, they lost my business . |

Table 4: With a trained *hier*-VAE-D model on the Yelp Review dataset, intermediate sentences are produced from linear transition between two points in the latent space.

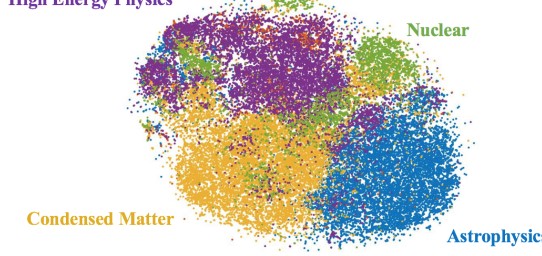

Figure 2: *t*-SNE visualization of the learned latent codes for arXiv abstracts.

**Visualization of the learned embeddings** To further prove the capability of our model to extract global features, we conducted an additional experiment to visualize the learned latent variable. Specifically, from arxiv dataset, we select the most frequent four classes/topics and re-train our hier-VAE-D model on the corresponding abstracts. The latent codes sampled from the learned inference network are visualized with *t*-SNE. As shown in Figure 2, each point indicates one paper abstract and the color of each point indicates the category it belongs to. The embeddings of the same label are indeed very close in the 2-D plot, while those with different labels are relatively farther away from each other. Additionally, the embeddings of the abstracts belonging to the *High Energy Physics* and *Nuclear* topics are meshed, which is expected since these two topics are semantically highly related.

| |
|---|
| we study the effect of disorder on the dynamics of a two-dimensional electron gas in a two-dimensional optical lattice , we show that the superfluid phase is a phase transition , we also show that , in the presence of a magnetic field , the vortex density is strongly enhanced . |
| in this work we study the dynamics of a colloidal suspension of frictionless , the capillary forces are driven by the UNK UNK , when the substrate is a thin film , the system is driven by a periodic potential , we also study the dynamics of the interface between the two different types of particles . |
| the problem of finding the largest partition function of a quantum system is equivalent to find the optimal value of the number of states , which is the most important example of the first order approximation of the ising model , the algorithm is based on the computation of the partition function , the results are compared with the results of the monte carlo simulations . |

Table 7: Generated samples from *hier*-VAE-D (trained on the arXiv abstract dataset).

Note that the model is trained in a totally unsupervised manner, and the inference network is able to extract meaningful global patterns from the input paragraph.

**Attribute Vector Arithmetic**   To further investigate the latent space's structure, we conduct an experiment to alter the sentiments of reviews with *attribute vector*. Specifically, on the Yelp Review dataset, we first obtain the sampled latent codes for all reviews with positive sentiment (among the entire training set), and calculate the corresponding mean latent vector. The mean latent vector for all negative reviews are computed in the same manner. The two vectors are then subtracted (i.e., positive mean vector minus negative mean vector) to obtain the "sentiment attribute vector". For evaluation, we randomly sample 1000 reviews with negative sentiment and add the "sentiment attribute vector" to their latent codes. The manipulated latent vectors are then fed to the hierarchical decoder to produce the transferred sentences (which should hopefully convey positive sentiment).

As shown in Table 8, the original sentences have been successfully manipulated to positive sentiment with the simple attribute vector operation. However, the specific contents of the reviews are not fully retained. One interesting future direction is to decouple the style and content of long-form texts to allow *content-preserving* attribute manipulation. We further employed a CNN sentiment classifier to evaluate the sentiment of manipulated sentences. The classifier is trained on the entire training set and achieves a test accuracy of $94.2\%$. With this pre-trained classifier, $83.4\%$ of the transferred reviews are judged to be positive-sentiment, indicating that "attribute vector arithmetic" consistently produces the intended manipulation of sentiment.

| | |
|---|---|
| **Original**: you have no idea how badly i want to like this place, they are incredibly vegetarian vegan friendly , i just haven t been impressed by anything i ve ordered there , even the chips and salsa aren t terribly good , i do like the bar they have great sangria but that s about it . | **Transferred**: this is definitely one of my favorite places to eat in vegas , they are very friendly and the food is always fresh, i highly recommend the pork belly , everything else is also very delicious, i do like the fact that they have a great selection of salads . |
| **Original**: my boyfriend and i are in our 20s , and have visited this place multiple times , after our visit yesterday , i don t think we ll be back , when we arrived we were greeted by a long line of people waiting to buy game cards . | **Transferred**: my boyfriend and i have been here twice , and have been to the one in gilbert several times too , since my first visit , i don t think i ve ever had a bad meal here , the servers were very friendly and helpful . |

Table 8: Sentiment transfer results with attribute vector arithmetic. More samples can be found in the supplementary material (Table 16).

**Human Evaluation**   We conducted human evaluation using Amazon Mechanical Turk to assess the coherence and non-redundancy of the texts generated from our models in comparison to the baselines, which is difficult to measure based on automated metrics. Given a pair of generated reviews, the judges are asked to select their preferences (no difference between the two reviews is also an option) according to the following four evaluation criteria: *fluency & grammar* (G), *consistency* (C), *non-redundancy* (N), and *overall* (O). Details of the evaluation are provided in the supplementary material.

| Model | G % | C % | N % | O % |
|---|---|---|---|---|
| *hier*-VAE | 52.0 | 55.0 | 53.7 | 60.0 |
| *flat*-VAE | 30.0 | 33.0 | 27.7 | 32.3 |
| *hier*-VAE | 75.3 | 86.0 | 76.7 | 86.0 |
| AAE | 13.3 | 10.3 | 15.0 | 12.0 |
| *flat*-VAE | 19.7 | 18.7 | 14.3 | 19.0 |
| Real data | 61.7 | 74.7 | 74.3 | 77.7 |
| *hier*-VAE | 28.0 | 26.3 | 25.0 | 30.3 |
| Real data | 48.6 | 58.7 | 49.0 | 61.3 |

Table 9: A Mechanical Turk blind heads-up evaluation between pairs of models trained on the Yelp Reviews dataset. From columns 2-5: Grammaticality(**G**), Consistency (**C**), Non-Redundancy(**N**), Overall (**O**).

As shown in Table 9, *hier*-VAE generates superior human-looking samples compared to *flat*-VAE on the Yelp Reviews dataset. Even though both models under-perfom when compared against the ground-truth real reviews, *hier*-VAE was rated higher in comparison to *flat*-VAE (raters find *hier*-VAE closer to human-generated than the *flat*-VAE) in all the criteria evaluation criteria. We further compare our methods against AAE (the same data preprocessing steps and hyperparameters are employed). The results show that *hier*-VAE again produces more grammatically-correct and semantically-coherent samples relative to the AAE baseline.

## 4.4    Conditional Paragraph Generation

| **Title**: *Magnetic quantum phase transitions of the antiferromagnetic - Heisenberg model* | **Title**: *Kalman Filtering With UNK Over Wireless UNK Channels* |
|---|---|
| We study the phase diagram of the model in the presence of a magnetic field, The model is based on the action of the Polyakov loop, We show that the model is consistent with the results of the first order perturbation theory. | The Kalman filter is a powerful tool for the analysis of quantum information, which is a key component of quantum information processing, However, the efficiency of the proposed scheme is not well understood . |

Table 10: Conditionally generated paper abstracts based upon a title (trained with the arXiv data).

We further evaluate the proposed VAE model on a conditional generation task. Specifically, we consider the task of generating abstract of a paper based on the corresponding title. The same arXiv dataset is utilized, where title and abstract are given as paired text sequences. The title is used as input of the inference network. For the generative network, instead of reconstructing the same input (*i.e.*, title), the paper abstract is employed as the target for decoding. We compare the *hier*-VAE-D model against *hier*-LM. We observe that the *hier*-VAE-D model achieves a test perplexity of $55.7$ (with a KL term of $2.57$), which is smaller that the test perplexity of *hier*-LM ($58.1$). This indicates that the information from the title has indeed been leveraged by the generative network to facilitate the decoding process. In table 10 we show two generated samples from the *hier*-VAE-D model.

## 4.5    Analysis

### 4.5.1    The architecture of encoder networks

To investigate the impact of encoder networks on the VAE's performance, we further conduct an ablation study, where the herarchical CNN encoder in the *hier*-VAE model is replaced with a flat CNN encoder or a hierarchical LSTM encoder, respectively. The corresponding results are shown in Table 11. It

| **Encoder Networks** | **NLL** | **KL** | **PPL** |
|---|---|---|---|
| flat CNN encoder | 164.6 | 2.3 | 50.2 |
| hierarchical LSTM encoder | 161.3 | 5.7 | 46.9 |
| hierarchical CNN encoder | 160.2 | 6.8 | 45.8 |

Table 11: Ablation study with different encoders.

can be observed that the model with a flat CNN encoder yields worst (largest) perplexity, suggesting that it is beneficial to make the encoder hierarchical. Additionally, hierarchical CNN encoder exhibits slightly better results than hierarchical LSTM encoder according to our experiments.

### 4.5.2    Mitigating "posterior collapse" issue

To resolve the "posterior collapse" issue of training textual VAEs, Park et al. (2018) also introduced a strategy called *utterance drop* (u.d). Specifically, they proposed to weaken the autoregressive power of hierarchical RNNs by dropping the utterance encoder vector with a certain probability. To investigate the effec-

| **Model** | **NLL** | **KL** | **PPL** |
|---|---|---|---|
| *hier*-VAE-S | 160.8 | 3.6 | 46.6 |
| *hier*-VAE-S (with u.d) | 161.3 | 5.6 | 47.1 |
| *hier*-VAE-D | 160.2 | 6.8 | 45.8 |

Table 12: Comparison with the *utterance drop* strategy.

tiveness of their method relative to our strategy of employing a hierarchy of latent variables, we further conduct a comparative study. Particularly, we utilize hier-VAE-S as the baseline model and apply the two strategies to it respectively. The corresponding results on language modeling (Yelp dataset) are shown in Table 12. Their u.d strategy indeed allows better usage of the latent variable (indicated by a larger KL divergence value). However, the NLL of the language model becomes even worse, possibly due to the weakening of the decoder during training (similar observations have also been reported in Table 2 of (Park et al., 2018)). In contrast, our hierarchical prior strategy yields larger KL terms as well as lower NNL value, indicating the advantage of our strategy to mitigate the "posterior collapse" issue.

## 5    Conclusion

We introduced a hierarchically-structured variational autoencoder for long text generation. A hierarchical LSTM generative network is employed, which models the semantic coherence at both the word- and sentence-levels. hierarchy of stochastic layers is further utilized, where the priors of the latent variables are learned from the data. Consequently, more informative latent codes are manifested, indicated by a larger KL loss term yet smaller variational lower bound. The generated samples from the proposed model also exhibit superior quality relative to those from several baseline methods (according to automatic metrics). Human evaluations further demonstrate that the samples from *hier*-VAE are less repetitive and more semantically-consistent.

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

# Supplementary Material for "Hierarchically-Structured Variational Autoencoder for Long Text Generation"

## A  DATASETS DETAILS

In the following, we provide details of dataset preprocessing as well as the experimental setups used in the experiments. For both Yelp Reviews and ArXiv Abstracts datasets, we truncate the original paragraph to the first five sentences (split by punctuation marks including *comma*, *period* and *point* symbols), where each sentence contains at most 25 words. Therefore, each paragraph has at most 125 words. We further remove those sentences that contain of less than 30 words. The statistics of both datasets are detailed in Table 13. Note that the average length of paragraphs considered here are much larger than previous generative models for text (Bowman et al., 2016; Yu et al., 2017; Hu et al., 2017; Zhang et al., 2017), since these works considered text sequences that contain only one sentence with at most twenty words.

| Dataset | Train | Test | Vocabulary | Average Length |
|---|---|---|---|---|
| Yelp Reviews | 244748 | 18401 | 12461 | 48 |
| ArXiv Abstracts | 504268 | 28016 | 32487 | 59 |

Table 13: Summary statistics for the datasets used in the generic text generation experiments.

## B    ADDITIONAL GENERATED SAMPLES FROM *hier*-VAE-D VS *flat*-VAE

We provide additional examples for the comparison between *hier*-VAE-D vs *flat*-VAE in Table 14, as a continuation of Table 3.

| *hier*-VAE | *flat*-VAE |
|---|---|
| i would give this place zero stars if i could , the guy who was working the front desk was rude and unprofessional , i have to say that i was in the wrong place , and i m not sure what i was thinking , this is not a good place to go to . | this is a great little restaurant in vegas , i had the shrimp scampi and my wife **had the shrimp scampi**, and my husband **had the shrimp scampi** , it was delicious , i **had the shrimp scampi** which was delicious and seasoned perfectly . |
| my wife and i went to this place for dinner , we were seated immediately , the food was good , i ordered the shrimp and grits , which was the best part of the meal . | **very good chinese food, very good chinese food**, the service was very slow, i guess that s what they were doing, very slow to get a quick meal. |
| we got a gift certificate from a store, we walked in and were greeted by a young lady who was very helpful and friendly, so we decided to get a cut, I was told that they would be ready in 15 minutes. | we go there for eakfast, i ve been here 3 times and it s always good, the hot dogs are delicious, and **the hot dogs are delicious**, i ve been there for eakfast and it is so good. |
| the place was packed, chicken was dry, tasted like a frozen hot chocolate, others were just so so, i wouldn t recommend this place. | do not go here, their food is terrible, they were very slow, in my opinion. |
| went today with my wife, and received a coupon for a free appetizer, we were not impressed, we both ordered the same thing, and **we were not impressed**. | the wynn is a great place to eat, the food was great and i had the linguine, and it was so good, i **had the linguine** and clams, ( i was so excited to try it ). |
| recently visited this place for the first time, i live in the area and have been looking for a good local place to eat, we stopped in for a quick bite and a few beers, always a nice place to sit and relax, wonderful and friendly staffs. | i came here for a quick bite before heading to a friend s recommendation, the place was packed, but the food was delicious, i am a fan of the place, and the place is **packed** with a lot of people. |
| best haircut i ve had in years, friendly staff and great service, he made sure that i was happy with my hair cut, just a little pricey but worth it, she is so nice and friendly. | had a great experience here today, the delivery was friendly and efficient and the food was good, i would recommend this place to anyone who will work in the future, will be back again. |
| great place to go for a date night, first time i went here, service is good, the staff is friendly, 5 stars for the food. | best place to get in vegas, ps the massage here is awesome, if you want to spend your money, then go there, ps **the massage is great**. |

Table 14: Samples randomly generated from *hier*-VAE-D and *flat*-VAE, which are both trained on the Yelp review dataset. The repetitive patterns within the generated reviews are highlighted.

## C    RETRIEVED CLOSEST TRAINING INSTANCES OF GENERATED SAMPLES (YELP REVIEWS DATASET)

We provide samples of retrieved instances from the Yelp Review training dataset which are closest to the generated samples. Table 15 shows the closest training samples of each generated Yelp review. The first column indicates the intermediate generated sentences produced from linear transition from a point $A$ to another point $B$ in the prior latent space. The second column on the right are the real sentences retrieved from the training set that are closest to the ones generated on the left (determined by BLEU-2 score). We can see that the retrieved training data is quite different from the generated samples, indicating that our model is indeed generating samples that it has never seen during training.

| | **Generated samples** | **Closest instance (in the training dataset)** |
|---|---|---|
| **A** | the service was great, the receptionist was very friendly and the place was clean, we waited for a while, and then our room was ready . | i ve only been here once myself , and i wasn t impressed , the service was great , staff was very friendly and helpful , we waited for nothing |
| • | same with all the other reviews, this place is a good place to eat, i came here with a group of friends for a birthday dinner, we were hungry and decided to try it, we were seated promptly. | i really love this place , red robin alone is a good place to eat , but the service here is great too not always easy to find , we were seated promptly , ought drinks promptly and our orders were on point . |
| • | this place is a little bit of a drive from the strip, my husband and i were looking for a place to eat, all the food was good, the only thing i didn t like was the sweet potato fries. | after a night of drinking , we were looking for a place to eat , the only place still open was the grad lux , its just like a cheesecake factory , the food was actually pretty good . |
| • | this is not a good place to go, the guy at the front desk was rude and unprofessional, it s a very small room, and the place was not clean. | the food is very good , the margaritas hit the spot , and the service is great , the atmosphere is a little cheesy but overall it s a great place to go . |
| • | service was poor, the food is terrible, when i asked for a refill on my drink, no one even acknowledged me, they are so rude and unprofessional. | disliked this place , the hostess was so rude , when i asked for a booth , i got attitude , a major . |
| **B** | how is this place still in business, the staff is rude, no one knows what they are doing, they lost my business . | i can t express how awful this store is , don t go to this location , drive to any other location , the staff is useless , no one knows what they are doing . |

Table 15: Using the *hier*-VAE-D model trained on the Yelp Review dataset, intermediate sentences are produced from linear transition between two points (**A** and **B**) in the prior latent space. Each sentence in the left panel is generated from a latent point on a linear path, and each sentence on the right is the closet sample to the left one within the entire training set (determined by BLEU-2 score).

## D    HUMAN EVALUATION SETUP AND DETAILS

Some properties of the generated paragraphs, such as (topic) coherence or non-redundancy, can not be easily measured by automated metrics. Therefore, we further conduct human evaluation based on 100 samples randomly generated by each model (the models are trained on the Yelp Reviews dataset for this evaluation). We consider *flat*-VAE, adversarial autoencoders (AAE) and real samples from the test set to compare with our proposed *hier*-VAE-D model. The same hyperparameters are employed for the different model variants to ensure fair comparison. We evaluate the quality of these generated samples with a blind heads-up comparison using Amazon Mechanical Turk. Given a pair of generated reviews, the judges are asked to select their preferences ("no difference between the two reviews" is also an option) according to the following 4 evaluation criteria: (1) *fluency & grammar*, the one that is more grammatically correct and fluent; (2) *consistency*, the one that depicts a sequence of topics and events that is more consistent; (3) *non-redundancy*, the one that is better at non-redundancy (if a review repeats itself, this can be taken into account); and (4) *overall*, the one that more effectively communicates reasonable content. These different criteria help to quantify the impact of the hierarchical structures employed in our model, while the non-redundancy and consistency metrics could be especially correlated with the model's plan-ahead abilities. The generated paragraphs are presented to the judges in a random order and they are not told the source of the samples. Each sample is rated by three judges and the results are averaged across all samples and judges.

# E    MORE SAMPLES ON ATTRIBUTE VECTOR ARITHMETIC

We provide more samples for sentiment manipulation, where we intend to alter sentiment of negative Yelp reviews with "attribute vector arithmetic", as a continuation of Table 8.

| | |
|---|---|
| **Original**: papa j s is expensive and inconsistent , the ambiance is nice but it doesn t justify the prices , there are better restaurants in carnegie . | **Transferred**: love the food , the prices are reasonable and the food is great , it s a great place to go for a quick bite . |
| **Original**: i had a lunch there once , the food is ok but it s on the pricy side , i don t think i will be back . | **Transferred**: i had a great time here , the food is great and the prices are reasonable , i ll be back . |
| **Original**: i have to say that i write this review with much regret , because i have always loved papa j s , but my recent experience there has changed my mind a bit , from the minute we were seated , we were greeted by a server that was clearly inexperienced and didn t know the menu . | **Transferred**: i have to say , the restaurant is a great place to go for a date , my girlfriend and i have been there a few times , on my last visit , we were greeted by a very friendly hostess . |
| **Original**: a friend recommended this to me , and i can t figure out why , the food was underwhelming and pricey , the service was fine , and the place looked nice . | **Transferred**: a friend of mine recommended this place , and i was so glad that i did try it , the service was great , and the food was delicious . |
| **Original**: this is a small , franchise owned location that caters to the low income in the area , selection is quite limited throughout the store with limited quantities on the shelf of the items they do carry , because of the area in which it is located , the store is not 24 hours as most giant eagle s seem to be . | **Transferred**: this is a great little shop, easy to navigate , and they are always open , their produce is always fresh , the store is clean and the staff is friendly . |

Table 16: Sentiment transfer results with attribute vector arithmetic.

