# OpenReview forum: "Hierarchically-Structured Variational Autoencoders for Long Text Generation"
_ICLR.cc/2019/Conference_

### Official Review · AnonReviewer2 · 2018-11-01

**Rating:** 7
**Confidence:** 4

**Review:**

This paper proposes a hierarchical variational autoencoder for modeling paragraphs. The model creates two levels of latent variables; one level of sentence-level latent variables and another single global latent. This avoids posterior collapse issues and the authors show convincing results on a few different applications to two datasets.

Overall, it is an impressive result to be able to convincingly model paragraphs with a useful global latent variable. Apart from some issues with confusing/incomplete notation (see below), my main criticism is that the authors fail to compare their approach to "A Hierarchical Latent Structure for Variational Conversation Modeling" by Park et al. As far as I can tell, the approaches are extremely similar, except that Park et al. may not learn the prior parameters and also use a hierarchical RNN encoder rather than a CNN (which may be irrelevant). They also are primarily interested in dialog generation, so the lower-level of their hierarchy models utterances in a conversation rather than sentences in general, but I don't see this as a major difference. I'd encourage the authors to compare to this and potentially use it as a baseline. More generally, it would have been nice to see more ablation experiments (e.g. convolutional vs. LSTM encoder). Finally, I know that space is tight, but other papers on global-latent-variable models tend to include more demonstrations that teh global variable is capturing meaningful information, e.g. with attribute vector arithmetic. The authors could include results of manipulating review sentiment via attribute vector arithmetic, for example.


Specific comments:

- "The Kullback-Leibler (KL) divergence term ... which can be written in closed-form (Kingma & Welling, 2013), encourages the approximate posterior distribution qφ(z|x) to be close to the multivariate Gaussian prior p(z)." The prior is not always taken to be a multivariate Gaussian. You should add a sentence stating that the VAE prior is often taken to be a diagonal-covariance Gaussian for convenience.
- 3.2 has a few things which are unclear. In the second paragraph, you define z as the sampled latent code which is fed through an MLP "to obtain the starting state of the sentence-level LSTM decoder". But then LSTM^{sent} appears to be fed z at every timestep. LSTM^{sent} is also not defined - am I to assume that its arguments are the previous state and current input, so that z is the input at every timestep? Also, you write "where h^s_0 is a vector of zeros" which makes it sound like the starting state of the sentence-level LSTM decoder is a vector of zeros, not the output of the MLP which takes z as input. In contrast, LSTM^{word} takes three arguments as input. Which are the "state" and which are the "input" to the LSTM?
- I don't see any description of your CNN encoder (only the LSTM decoder in section 3.2, 3.3 only covers the hierarchy of latent variables, not the CNN architecture). What is its structure? Figure 1 shows a CNN encoder generating lower-level sentence embeddings and a high-level global embedding. How are those computed? It is briefly mentioned in 4.1 under "Datasets" but this seems insufficient.
- p_\theta(x | z) is defined as the generating distribution, but also as a joint distribution of z_1 and z_2. Unless I am missing something I think you are overloading the notation for p_\theta.
- I don't think enough information is given about the AAE and ARAE baselines. Are they the same as the flat-VAE, except with the KL term replaced by the an adversarial divergence between the prior and approximate posterior?

---

> ### Author Response · Authors · 2018-11-17
> **Thanks for your kind and helpful comments**
>
> Thanks for taking the time to provide such thorough comments and detailed feedback.
>
> > my main criticism is that the authors fail to compare their approach to "A Hierarchical Latent Structure for Variational Conversation Modeling" by Park et al.
>
> Thanks for referring to this interesting paper. The VHCR model (as denoted in their paper) indeed bears close resemblance to our proposed method. However, there are two key differences that make our work unique:
>
> 1) To allow the model to make more use of the latent variable, this reference proposes to employ an ‘utterance drop’ strategy. Their goal is to weaken the autoregressive power of hierarchical RNNs by dropping the utterance encoder vector with a certain probability. Although the KL divergence term tends to get larger with this modification, the modeling capacity of the LSTM decoder may be sacrificed. Instead, we try to resolve the same issue from a different perspective (without weakening the decoder during training). Specifically, we propose to improve the flexibility and expressiveness of the prior distribution, by leveraging a hierarchy of latent variables. This setup endows the encoder/inference networks with stronger ability to extract high-level (global) features of a paragraph. Notably, the model proposed in [1] shares the same high-level idea of making the inference networks more expressive to resolve the ‘’posterior collapse’’ problem.
>
> To compare the effectiveness of the two different strategies (to mitigate the `posterior collapse’ issue), we experiment the ‘utterance drop’ (u.d) method based upon our hier-VAE-S model on the Yelp dataset (note that to allow fair comparison, we use hier-VAE-S as the baseline to evaluate the two strategies). The corresponding language modeling results are shown as below:
>
> 			                    NLL        KL        PPL
> hier-VAE-S: 		           160.8      3.6       46.6
> hier-VAE-S (with u.d):     161.3  	  5.6       47.1
> hier-VAE-D: 		           160.2      6.8       45.8
>
> As shown above, their u.d strategy allows better usage of the latent variable (indicated by a larger KL divergence value). However, the NLL of the language model becomes even worse with the u.d method, possibly due to the weakening of the decoder during training (similar observations have also been shown in Table 2 of the VHCR paper). In contrast, our ‘hierarchical prior’ strategy yields larger KL terms as well as lower NNL value, indicating the advantage of our strategy to mitigate the ‘posterior collapse’ issue.
>
> 2) The previous VHCR model considers a multi-turn dialogue generation scenario, where the dialog context is provided at each time step (in terms of the higher-level LSTM) and the model seeks to generate the corresponding response conditioned on the context. What is different in our hier-VAE model is that, we are interested in generating long and coherent units merely conditioned on the latent variable (no additional context information is provided to the decoder). As a result, the underlying data distribution of the entire paragraph is captured in the bottom-level latent variable. While in their setup, the responses are modeled/generated conditioned on both the latent variables and the contexts. In this sense, the problem we are trying to tackle is relatively more challenging.
>
> > More generally, it would have been nice to see more ablation experiments (e.g. convolutional vs. LSTM encoder)
>
> At the initial stage of this project, we found that the hierarchical CNN encoder employed here is important for the VAE model to work well (relative to a flat CNN encoder). Based on reviewer’s suggestion, we re-ran the language modeling experiments with a flat CNN encoder and a hierarchical LSTM encoder (on the Yelp dataset). The results are shown below:
>
> 					                          NLL        KL        PPL
> Flat CNN encoder:			         164.6      2.3       50.2
> Hierarchical LSTM encoder:		 161.3	 5.7       46.9
> Hierarchical CNN encoder:                 160.2      6.8       45.8
>
> It can be observed that the model with a flat CNN encoder yields worst (largest) perplexity, suggesting that it is beneficial to make the encoder hierarchical. Additionally, hierarchical CNN encoder exhibits slightly better results than hierarchical LSTM encoder according to our experiments.

---

> > ### Author Response · Authors · 2018-11-17
> > **Continue due to the character limit**
> >
> > > Finally, I know that space is tight, but other papers on global-latent-variable models tend to include more demonstrations that the global variable is capturing meaningful information, e.g. with attribute vector arithmetic.
> >
> > We share the same intuition with the reviewer that exploring what information has been meaningfully abstracted in the (global) latent variable is valuable to understand the strengths of our proposed model. We would like to mention that in our original submission, we included a section in the experiments where we reported results to measure the continuity of the learned latent space. This measure is commonly used in latent-variable models to evaluate how smooth the learned VAE latent space is. Our results demonstrated that the generated samples are syntactically and semantically reasonable. We even found that the generated sentences, along a linear trajectory, gradually transit from positive to negative sentiment (as shown in Table 4). Having said that, we followed reviewer's suggestion and are currently running additional experiments to manipulate the review sentiment via attribute vector arithmetic. We will update with the corresponding results shortly.
> >
> > Furthermore, we’ve just reported the results of our new experiment to visualize the learned global latent variable by plotting the corresponding t-SNE embeddings on the arXiv dataset (please refer to our response to Reviewer 1 for additional information).
> >
> > Below, please also find our responses to the specific comments of the reviewer:
> >
> > [VAE prior] Thanks for pointing this out. We have revised this sentence according to your suggestion.
> >
> > [Feeding z to LSTM] We apologize for the confusion. As to feeding the latent codes z to LSTM, we follow [2] and use z to infer the initial state of LSTM and feed it to every step of LSTM as well. That said, z is employed as the input to the sentence-level LSTM (higher level decoder) at every time step. For the word-level LSTM, the plan vectors are utilized by the decoder in a similar manner. Different from the higher-level LSTM, the input to the word-level LSTM at each step is the concatenation of z and the word embedding of the previous token. We have revised the description of this part in Section 3.2 to avoid any confusion.
> >
> > [CNN encoder structure] Due to space limit, the specific structure of CNN encoder is described in the supplementary materials. To make it more clear, we added a ''Model specification‘’ section in the revised version, in the beginning of ''Experiments‘’ part, detailing the encoder structure..
> >
> > [Notation] \theta is defined as the parameters of the whole generative network. The individual network that parametrize p(x|z_1) and p(z_1|z_2) are both part of the generative networks. Therefore, we use \theta to denote the parameters of both distributions.
> >
> > [AAE and ARAE baselines] Yes, the setups of the two baseline methods are the same as flat-VAE, except that adversarial divergence, instead of the KL divergence, is employed to match the prior and posterior distributions.
> >
> > [1] Kim, Y., Wiseman, S., Miller, A.C., Sontag, D.A., & Rush, A.M. (2018). Semi-Amortized Variational Autoencoders. ICML.
> > [2] Bowman, S.R., Vilnis, L., Vinyals, O., Dai, A.M., Józefowicz, R., & Bengio, S. (2016). Generating Sentences from a Continuous Space. CoNLL.

---

> > > ### Comment · AnonReviewer2 · 2018-11-20
> > > **Response**
> > >
> > > Thanks for your detailed consideration of my comments. While I agree that a big difference between your work and that of Park et al. is their use of "utterance drop", I'm not sure I agree that the hierarchical structure of the two models is significantly different. You wrote
> > >
> > > > Specifically, we propose to improve the flexibility and expressiveness of the prior distribution, by leveraging a hierarchy of latent variables. This setup endows the encoder/inference networks with stronger ability to extract high-level (global) features of a paragraph. Notably, the model proposed in [1] shares the same high-level idea of making the inference networks more expressive to resolve the ‘’posterior collapse’’ problem.
> > >
> > > It appears they also do this - see equations (24) and (25) in Park et al., where the utterance variables are treated as stochastic and are dependent on the global conversation variable. If there is indeed an important different in the structure of your latent variables, you need to make that absolutely clear in your updated draft. Otherwise, it seems that the only difference is their use of "utterance drop" and the difference in applications, neither of which are terribly significant. Looking at your updated draft, it appears you have only mentioned Park et al. in pointing out their use of utterance drop. I think you need to flesh out this comparison.
> > >
> > > With the revisions you made, experiments with "attribute vector arithmetic", and some additional words comparing your work to Park et al.,  I would be happy to raise my score.

---

> > > > ### Author Response · Authors · 2018-11-23
> > > > **“Attribute vector arithmetic” results & comparison with related work**
> > > >
> > > > Thanks for your detailed response. According to your valuable feedback, we have made two further revisions on our manuscript:
> > > >
> > > > 1) The ‘’attribute vector arithmetic’’ experiment: on the Yelp Review dataset, we first obtain the sampled latent codes for all reviews with positive sentiment (among the entire training set), and calculate the corresponding mean latent vector. The mean latent vector for all negative reviews are computed in the same manner. The two vectors are then subtracted (i.e., positive mean vector minus negative mean vector) to obtain the ‘’sentiment attribute vector’’. For evaluation, we randomly sample 1000 reviews with negative sentiment and add the ‘’sentiment attribute vector’’ to their latent codes. The manipulated latent vectors are then fed to the hierarchical decoder to produce the transferred sentences (which should hopefully convey positive sentiment). Below are some transferred results:
> > > >
> > > > >>>>>>>>>>>>>>>>>>>>>>>>>>>>>>>>>>>>>>>>>>>>>>>>>>>>>>>>>>>>>>>>>>>>>>>>>>>>>>>>>>>>>>>>>>>>>>>>>>>>>>>>>>>>>>>>>>>>>>>>>>>>>>>>
> > > > Original: you have no idea how badly i want to like this place, they are incredibly vegetarian vegan friendly , i just haven t been impressed by anything i ve ordered there , even the chips and salsa aren t terribly good , i do like the bar they have great sangria but that s about it .
> > > >
> > > > Transferred: this is definitely one of my favorite places to eat in vegas , they are very friendly and the food is always fresh , i highly recommend the pork belly , everything else is also very delicious , i do like the fact that they have a great selection of salads .
> > > > ---------------------------------------------------------------------------------------------------------------------------------------------------------------------------------------------------------------
> > > > Original: papa j s is expensive and inconsistent , the ambiance is nice but it doesn t justify the prices , there are better restaurants in carnegie .
> > > >
> > > > Transferred: love the food , the prices are reasonable and the food is great , it s a great place to go for a quick bite .
> > > > ----------------------------------------------------------------------------------------------------------------------------------------------------------------------------------------------------------------
> > > > Original: my boyfriend and i are in our 20s , and have visited this place multiple times , after our visit yesterday , i don t think we ll be back , when we arrived we were greeted by a long line of people waiting to buy game cards .
> > > >
> > > > Transferred: my boyfriend and i have been here twice , and have been to the one in gilbert several times too , since my first visit , i don t think i ve ever had a bad meal here , the servers were very friendly and helpful .
> > > > --------------------------------------------------------------------------------------------------------------------------------------------------------------------------------------------------------------
> > > > Original: a friend recommended this to me , and i can t figure out why , the food was underwhelming and pricey , the service was fine, and the place looked nice .
> > > >
> > > > Transferred: a friend of mine recommended this place , and i was so glad that i did try it , the service was great , and the food was delicious .
> > > > >>>>>>>>>>>>>>>>>>>>>>>>>>>>>>>>>>>>>>>>>>>>>>>>>>>>>>>>>>>>>>>>>>>>>>>>>>>>>>>>>>>>>>>>>>>>>>>>>>>>>>>>>>>>>>>>>>>>>>>>>>>>>>>>
> > > >
> > > > As shown above, the original sentences have been successfully manipulated to positive sentiment with the simple attribute vector operation. Notably, the general contents of the original sentences are mostly retained, even though their sentiments are altered. We further employed a CNN sentiment classifier to evaluate the sentiment of manipulated sentences. The classifier is trained on the entire training set and achieves a test accuracy of 94.2%. With this pre-trained classifier, 83.4% of the transferred reviews are judged to be positive-sentiment, indicating that ‘’attribute vector arithmetic’’ method consistently produces the intended manipulation of sentiment. We have included the results and discussion in our revised manuscript.

---

> > > > > ### Author Response · Authors · 2018-11-23
> > > > > **Continue due to the character limit**
> > > > >
> > > > > 2) We agree with the reviewer that a clear comparison with the VHCR model in Park et al. is important. In the revised manuscript, we particularly discuss the differences between our method and the VHCR model in the “Hierarchical structures in NLP” part of section 2. Specifically, our approach is unique from the following perspectives: (i) both latent variables in our hier-VAE are designed to contain global information. More importantly, although the local/utterance variables are generated from the global latent variable in VHCR, the priors for the two sets of latent variables are both fixed as standard diagonal-covariance Gaussians. In contrast, the prior of the bottom-level latent variable in our model is learned from the data (thus more flexible relative to a fixed prior), which exhibits promising results in terms of mitigating the “posterior collapse” issue (see Table 2); (ii) in hier-VAE, the underlying data distribution of the entire paragraph is captured in the bottom-level latent variable. While in the setup of VHCR, the responses are modeled/generated conditioned on both the latent variables and the contexts. Therefore, the (global) latent variable learned by our model should contain more information.

---

> > > > > > ### Comment · AnonReviewer2 · 2018-11-24
> > > > > > **Re: “Attribute vector arithmetic” results & comparison with related work**
> > > > > >
> > > > > > Thanks for doing the attribute vector experiments and expanding your discussion of VHCR.
> > > > > >
> > > > > > > As shown above, the original sentences have been successfully manipulated to positive sentiment with the simple attribute vector operation. Notably, the general contents of the original sentences are mostly retained, even though their sentiments are altered.
> > > > > >
> > > > > > To be frank, I totally disagree. Most of the sequences change both in content and in sentiment. For example, in the first example the original talks about being vegetarian/vegan friendly, chips and salsa, and sangria; the transferred about pork belly and salad. The only real instance of keeping content similar is "my boyfriend and i" appearing in the second-to-last example, but otherwise the content is totally different - the original is talking about buying game cards; the transferred is talking about a restaurant. So I don't think you can claim your model has decoupled style and content in any meaningful way. That being said, I think it's important to include this kind of result in your paper, mostly to motivate future work.
> > > > > >
> > > > > > I appreciate the work you've done to improve the paper. I've raised my score to a 7.

---

> > > > > > > ### Author Response · Authors · 2018-11-25
> > > > > > > **Re: Re: “Attribute vector arithmetic” results & comparison with related work**
> > > > > > >
> > > > > > > Thanks for your consideration of our efforts to improve the manuscript. We agree that this claim (i.e., "the general contents of the original sentences are mostly retained") is not rigorous enough. In this regard, we have refined this part and updated a revised manuscript.

---

### Official Review · AnonReviewer1 · 2018-11-04
**Nice samples but lack of comparison to existing hierarchical VAEs**

**Rating:** 5
**Confidence:** 4

**Review:**

This paper proposes using a hierarchical VAE to text generation to solve the two problems of long text generation and mode collapse where diversity in generated text is lost.

The paper does this by decoding the latent variable into sentence level latent codes that are then decoded into each sentence. The paper shows convincing results on perplexity, N-gram based and human qualitative evaluation.

The paper is well written though some parts are confusing. For example, equation 4 refers to q as the prior distribution but this seems like it's the posterior distribution as it is described just below equation 5. p(z_1|z_2) is also not well defined. It would be clearer to specify the full algorithm in the paper.

The work also mentions that words are generated for each sentence until the _END token is generated. Is this token always generated? What happens to a sentence if that token is not generated?

The novelty of this paper is questionable given the significant amount of existing work in hierarchical VAEs. It's also unclear why a more direct comparison can't be made with Serban et. al in terms of language generation quality and perplexity. If a downstream model is only able to make use of one latent variable, can't multiple variables simply be averaged?

It's also unclear how this work is novel with regards to the works below.

Hierarchical Variational Autoencoders for Music
Roberts, et. al
NIPS 2017 creativity workshop
This seems to have a similar hierarchical structure where there is an initial 16 step decoder that decodes the latent code for the lower level note level LSTMs to use during generation.

Unsupervised Learning of Disentangled and Interpretable Representations from Sequential Data
Hsu, et. al
NIPS 2017
This proposes a factorized hierarchical variational autoencoder which also has a double latent variable hierarchical structure, one that is conditional on the other.

Minor comments
- Typo in page 3 under Hierarchical structures in NLP: characters "from" a word
- Typo above section 4.3: hierarhical

=== After rebuttal ===
Thanks for the response.

I believe that Reviewer2's criticism about the similarity to Park et. al isn't sufficiently addressed by the authors. Even if the hierarchical structure is different it's unclear whether this alternative structure is superior to Park et. al. There appears to be no evidence that the latent variables contain more global information relative to VHCR (Park et. al). These claims aren't tested and the results in the paper aren't comparable since the authors don't evaluate on the same datasets as Park et. al.

In general, I think the claims of a superior hierarchical structure to models such as the factorized hierarchical VAE paper needed to be tested to show evidence of a more powerful representation for hier-VAE.

I will keep my score.

---

> ### Author Response · Authors · 2018-11-17
> **Thanks for the valuable, critical feedback**
>
> We would like to thank the reviewer for the detailed comments and suggestions for the manuscript.
>
> > The paper is well written though some parts are confusing. For example, equation 4 refers to q as the prior distribution but this seems like it's the posterior distribution as it is described just below equation 5. p(z_1|z_2) is also not well defined. It would be clearer to specify the full algorithm in the paper.
>
> Thanks for pointing out this confusing typo. In the revised version, we have corrected ‘prior’ to ‘posterior’ while describing equation 4. As to the specific configuration of p(z_1|z_2), it is actually described in the supplementary materials (due to the space limit). To avoid any confusions here, we added a 'Model specification’ section, at the beginning of the 'Experiments' part in the revised version, to illustrate these details and hyperparameter choices.
>
> > The work also mentions that words are generated for each sentence until the _END token is generated. Is this token always generated? What happens to a sentence if that token is not generated?
>
> The reviewer brings up a good point about the end_token generation. In our training data, all sentences have an _END token (at the end of each sentence), and the word-level LSTM is able to learn that there will be an _END token after each sentence and generate that token.  We should also note that, we constrained the model to generate a maximum number of words for each sentence. We use maximumly 20 words for the Yelp reviews dataset and 25 for the arXiv dataset. This part has been made more clear in the revised version.
>
> > It's also unclear how this work is novel with regards to the works below.
>
> We agree with the reviewer that there are similarities between our model and the VHRED model proposed by Serban et. al., however our model is quite different in terms of the architecture and motivation. Firstly, the VHRED model infers a latent variable for each context/utterance, and the context is provided to the decoder while generating each response. In contrast, our model leverages a global latent variable to model the entire paragraph without any additional inputs (no contexts are provided). As a result, the latent variables in our model are designed to abstract globally meaningful features from the entire paragraph.
>
> Secondly, we have leveraged a hierarchy of latent variables to mitigate the ‘posterior collapse’ issue, which has given rise to promising empirical results. Specifically, the generative network has made better use of the latent variable information, indicated by a larger KL term (please refer to our response to Reviewer 2 for additional information). This strategy has not been employed and discussed in the ‘Hierarchical Variational Autoencoders for Music’ paper either.
>
> To further prove that our model can extract globally informative features, we conducted an additional experiment, in the revised paper, to visualize the learned latent variable. Specifically, from the arXiv dataset, we select the most frequent four classes/topics and re-train our hier-VAE-D model on the corresponding abstracts. We plot the bottom-level latent variable for our hier-VAE-D with t-SNE (which is supposed to contain the global features). The result is shown in Figure 2 of the revised manuscript. It can be observed that the latent codes of paragraphs from the same topic are indeed grouped together in the embedding space, indicating that the latent variables has encoded high-level features from the input paragraph.
>
> As to the factorized hierarchical VAE paper, although they utilize the hierarchical nature of sequential data, their VAE model only takes a sub-sequence (segment) of the entire sequence as the input (as illustrated in Figure 3 of the factorized hierarchical VAE paper). Therefore, their model can only be used to modify the style/attribute of a sub-sequence (short sentence), rather than generate/sample an entire sequence (long-form document).
>
> > Minor comments
>
> Thanks for pointing the typos out. We have corrected them accordingly in the revised paper.

---

### Official Review · AnonReviewer3 · 2018-11-06
**Nice experiments, but limited novelty**

**Rating:** 5
**Confidence:** 4

**Review:**

This paper proposed a hierarchical generative model for generating long text. The authors use a hierarchical LSTM decoder to first generate sentence-level representations; then based on the representation of each sentence, a word-level LSTM decoder is utilized to generate a sequence of words in this sentence. In addition, they use multiple layers of latent variables to address the posterior collapse issue.
The paper studies an important problem and the authors performed extensive experiments.


My major concern is about the novelty of this paper.
Hierarchical LSTM for generating long txt has been widely studied. For example, in the following works:
Li, Jiwei, Minh-Thang Luong, and Dan Jurafsky. "A hierarchical neural autoencoder for paragraphs and documents." arXiv preprint arXiv:1506.01057 (2015).
Hierarchical LSTM for Sign Language Translation, AAAI, 2018.

Placing hierarchical latent variables in VAE  is also investigated before.
For example, in
Zhao, Shengjia, Jiaming Song, and Stefano Ermon. "Infovae: Information maximizing variational autoencoders." arXiv preprint arXiv:1706.02262 (2017).  With some adaption from image domain to text domain
Serban, Iulian Vlad, et al. "A Hierarchical Latent Variable Encoder-Decoder Model for Generating Dialogues." AAAI. 2017.

The author combines this two ideas together, which is incremental in terms of novelty.


In the writing of section 3.2, the authors should clearly cite the previous works on hierarchical LSTM and acknowledge that this is not the contributions of this paper. Under the current writing, for unfamiliarized readers, it sounds like this is proposed by the authors of this paper, which is not the case.

The notations of this paper is confusing, which hinders its readbility.
For example, in equation 5, the distribution is parameterized by theta.
In equation 6, p(x|z) is also parametrized by theta.

In the experiments, I'd like to see a comparison with the following works.
I suggest the authors to compare with the following works.

Fan, Angela, Mike Lewis, and Yann Dauphin. "Hierarchical Neural Story Generation." ACL (2018).

Ghosh, S., Vinyals, O., Strope, B., Roy, S., Dean, T., & Heck, L. (2016). Contextual LSTM: A Step towards Hierarchical Language Modeling.

Zhao, Shengjia, Jiaming Song, and Stefano Ermon. "Infovae: Information maximizing variational autoencoders." arXiv preprint arXiv:1706.02262 (2017).  With some adaption from image domain to text domain

---

### Meta-Review · Area_Chair1 · 2018-12-13
**Interesting latent variable model, borderline paper due to experimental execution and novelty**

**Confidence:** 4
**Recommendation:** Reject

**Metareview:**

Strengths: Interesting work on using latent variables for generating long text sequences.
The paper shows convincing results on perplexity, N-gram based and human qualitative evaluation.

Weaknesses: More extensive comparisons with hierarchical VAEs and the approach in Serban et. al in terms of language generation quality and perplexity would have been helpful. Another point of reference for which additional comparisons were desired was: "A Hierarchical Latent Structure for Variational Conversation Modeling" by Park et al. Some additional substantive experiments were added during the discussion period.

Contention: Authors differentiated their work from Park et al. and the reviewer bringing this work up ended up upgrading their score to a 7. The other reviewers kept their scores at 5.

Consensus: The positive reviewer raised their score to a 7 through the author rebuttal and discussion period.  One negative reviewer was not responsive, but the other reviewer giving a 5 asserts that they maintain their position. The AC recommends rejection. Situating this work with respect to other prior work and properly comparing with it seems to be the contentious issue. Authors are encouraged to revise and re-submit elsewhere.